# Human Serum and Salivary Metabolomes: Diversity and Closeness

**DOI:** 10.3390/ijms242316603

**Published:** 2023-11-22

**Authors:** Elena Ferrari, Mariana Gallo, Alberto Spisni, Rita Antonelli, Marco Meleti, Thelma A. Pertinhez

**Affiliations:** 1Laboratory of Biochemistry and Metabolomics, Department of Medicine and Surgery, University of Parma, 43126 Parma, Italy; elena.ferrari@unipr.it (E.F.); alberto.spisni@unipr.it (A.S.); thelma.deaguiarpertinhez@unipr.it (T.A.P.); 2Centro Universitario Odontoiatria, University of Parma, 43126 Parma, Italy; rita.antonelli@unipr.it (R.A.); marco.meleti@unipr.it (M.M.)

**Keywords:** salivary metabolites, serum metabolites, metabolite profile, parotid gland, submandibular gland, sublingual glands, whole saliva

## Abstract

Saliva, which contains molecular information that may reflect an individual’s health status, has become a valuable tool for discovering biomarkers of oral and general diseases. Due to the high vascularization of the salivary glands, there is a molecular exchange between blood and saliva. However, the composition of saliva is complex and influenced by multiple factors. This study aimed to investigate the possible relationships between the salivary and serum metabolomes to gain a comprehensive view of the metabolic phenotype under physiological conditions. Using ^1^H-NMR spectroscopy, we obtained the serum metabolite profiles of 20 healthy young individuals and compared them with the metabolomes of parotid, submandibular/sublingual, and whole-saliva samples collected concurrently from the same individuals using multivariate and univariate statistical analysis. Our results show that serum is more concentrated and less variable for most of the shared metabolites than the three saliva types. While we found moderate to strong correlations between serum and saliva concentrations of specific metabolites, saliva is not simply an ultrafiltrate of blood. The intense oral metabolism prevents very strong correlations between serum and salivary concentrations. This study contributes to a better understanding of salivary metabolic composition, which is crucial for utilizing saliva in laboratory diagnostics.

## 1. Introduction

Whole saliva (WS) contains molecular information that may reflect the health status of an individual. It is a complex water mixture composed of organic and inorganic secretions of major and minor salivary glands, gingival crevicular fluid, suspended bacteria, desquamated cells, and food debris [1,2,3]. Its production is significantly affected by circadian rhythm, age, hydration status, physical exercise, oral hygiene, and food consumption [1,4]. In addition, blood-derived molecules may enter the highly vascularized salivary glands via transcellular and paracellular routes, affecting the biochemical composition of saliva [2,5,6]. In the oral cavity, the metabolite composition of whole saliva is further modified by the metabolism of the resident oral microflora, desquamated host cells, and immune cells [7]. As a result, oral fluids contain molecular information that may reflect the health status of an individual.

### 1.1. Salivary Glands

The major glands that bilaterally secrete saliva are the parotid, submandibular, and sublingual glands (Figure 1A). Under the control of the autonomic nervous system, these major salivary glands create about 90% of the saliva, with the remaining saliva, which is produced by 600 to 1000 minor salivary glands, found mainly on the lips, buccal mucosa, palate, and tongue [8]. 

Salivary glands are composed of specialized epithelial cells that form acini, the glandular end portions that initiate the secretion of saliva (Figure 1A), and branched ducts through which saliva enters the oral cavity. The acini can be serous, mucous, or a combination of both, determining the secretion type they produce. While a serous acinus cell secretes a watery salivary fluid rich in proteins, a mucous one produces a viscous secretion rich in mucins [2]. 

### 1.2. From Capillary Blood to Whole Saliva

A dense network of capillaries surrounds the ductal system, while a less intense vascularization supplies the terminal acini. When stimulated by parasympathetic vasodilator fibers, blood flow increases up to 20-fold and produces the flux of a solution, containing inorganic and small organic substances (Figure 1B), toward the gland lumen via (a) ultrafiltration through the narrow junctions between acinar cells, (b) intracellular passive diffusion according to a concentration gradient, or (c) energy-dependent transport against a concentration gradient [4,5,11,12,13]. 

### 1.3. Salivary Metabolomics

Over the last two decades, knowledge about saliva and its homeostasis has increased to such an extent that a new term, salivaomics, has been coined to describe the reservoir of information present in WS, referring to the subject’s genome, epigenome, transcriptome, proteome, metabolome, and microbiome [14,15]. 

Among the organic macro-components, salivary proteins have been extensively investigated, revealing their glandular (e.g., α-amylase, mucins, histatins, cystatins, proline-rich proteins, statherins, lactoferrin, and lysozyme) and plasma (e.g., albumin, secretory immunoglobulin A, and transferrin) origin [1]. The presence of plasma proteins in saliva represents a source of new markers of diseases, offering a diagnostic alternative to specific blood tests [1,16,17,18].

The comprehensive analysis of metabolites in a salivary specimen has been employed (a) as a source of biomarkers for oral and general disorders, (b) to study changes in saliva composition with exercise performance, and (c) to correlate the salivary metabolome with oral microbiome [3,19,20,21]. Our recent investigations of the salivary metabolome of whole, parotid, and submandibular/sublingual saliva in healthy and young individuals [22,23] prompted us to explore the relationships between the salivary and serum metabolomes to provide a comprehensive view of the metabolic phenotype in physiological conditions. Here, we present (a) the serum metabolite composition of young and healthy subjects obtained using ^1^H-NMR spectroscopy and (b) a comparison of their serum and salivary metabolite profiles. 

## 2. Results

### 2.1. Sample Collection and Metabolite Profiling 

Serum, parotid saliva (PS), submandibular/sublingual saliva (SM/SLS), and WS samples were collected from each subject according to a previously established protocol [22]. 

Metabolomic analysis based on ^1^H-NMR spectroscopy allowed the identification of 43 metabolites in the serum of all study participants. Appendix A reports their mean concentrations and standard deviations. 

Each participant released the three salivary samples (5.4 mL) at different rates (Figure 2, Appendix A). The mean sample collection rates of PS, SM/SLS, and WS were consistent with the salivary flow rates presented in the literature [24] and corresponded to an approximate ratio of 1:2:4.5, suggesting a comparable correlation between their secretion rates.

A list of the metabolites’ mean concentrations in the three types of saliva and an analysis of the resulting salivary profiles have already been published [22,23]. The obtained salivary profiles, rearranged according to chemical category, are presented in Appendix A.

### 2.2. Comparison of Serum and Salivary Metabolite Profiles 

The number of metabolites detected in all sample types was computed and plotted in a Venn diagram (Figure 3). In this representation, the area covered by the three circles highlights the core of 31 shared metabolites identified in all matrices (Appendix A). This number drops to 27 when separating the glandular saliva profile (green circle) into two distinct profiles, i.e., PS and SM/SLS (27 = 31 − 4 metabolites common to PS and SM/SLS). The overlapping section of each pair of circles shows the number of shared metabolites. Metabolites that are unique to each sample type are listed in Figure 3.

We performed a partial least-squares–discriminant analysis (PLS-DA) using the NMR-derived concentrations of the metabolites shared by all sample matrices (*n* = 27). The PLS-DA score plot presented in Figure 4A evidences a net clustering for serum, PS, and SM/SLS samples, whereas WS samples exhibit a notable dispersion. The model denotes a good separation by component 1 between serum and all salivary samples; metabolite concentration differences do not appreciably distinguish the three salivary samples. Figure 4B reports the Variable Importance in Projection (VIP) scores of the top 15 discriminant metabolites in order of importance in the PLS-DA model, according to the variance explained by component 1. Differences in Acetate, Glucose, and Lactate concentrations, with VIP scores >2, are the most important in generating cluster separation. 

Upon excluding the most discriminant metabolites, Acetate, Glucose, and Lactate, the PLS-DA model still allowed for some separation between serum and the salivary samples by component 2. In this case, the most discriminant metabolites (1 < VIP score < 4) were Urea, Glutamine, Alanine, and Taurine. Serum exhibited higher levels of Glutamine and Alanine than the salivary samples. The role of these amino acids as gluconeogenic precursors, being mainly released by muscle tissue during fasting, may explain these findings. However, Taurine’s prevalence in PS suggests a role of this metabolite in salivary gland function.

### 2.3. Comparison of Salivary and Serum Metabolite Concentrations

#### 2.3.1. Metabolites Common to Serum and Saliva Collected from Parotid or Submandibular/Sublingual Glands

We used the Mann–Whitney test to compare the metabolite composition of serum and PS or SM/SLS. The concentration distribution significantly differed for all metabolites, except for Choline in serum and PS and Creatine, Glutamate, and Urea in serum and SM/SLS. 

We performed a correlation analysis for metabolite concentrations. For most of them, the correlation was negligible when comparing serum to PS or SM/SLS. Based on Pearson analysis, the highest degree of correlation was obtained with 2-Hydroxyisovalerate, 2-Hydroxybutyrate, and 3-Hydroxybutirate (0.5 ≤ r < 0.7) when comparing serum to PS and with Phenylalanine (r = 0.5) and Pyruvate (r = −0.5) when comparing serum to SM/SLS. In serum vs. PS, Spearman correlation analysis confirmed all the strong/moderate correlations found in Pearson analysis and assigned a moderate correlation (|r| ≈ 0.5) to Creatine and Lactate. In serum vs. SM/SLS, Spearman correlation analysis confirmed the moderate negative correlation of Pyruvate found in Pearson analysis and assigned a moderately positive correlation (r ≈ 0.5) to 3-Hydroxybutyrate, Creatine, and Creatinine. 

We calculated the concentration ratios of “serum/saliva” for each metabolite present in serum and saliva. These ratios were averaged across the 20 study participants and are referred to as “mean fold change”, a parameter that expresses the metabolite concentration difference between serum and saliva samples. Figure 5 depicts those metabolites’ mean fold change as a function of their correlation coefficients. In both graphs, we have labeled the metabolites with mean fold changes <0.5 or >3, likely reflecting significant differences: for values lower than 0.5, we expected the metabolite salivary concentration to be higher than in serum; for values higher than 3, we expected the metabolite salivary concentration to be lower than in serum.

In the comparison of serum vs. PS (Figure 5A), among the metabolites which exhibited a mean fold change >3 or <0.5, only Proline and Pyruvate resulted in a limited number of cases where the concentration ratio was reversed. In fact, their serum concentrations were lower than the corresponding salivary concentrations in some subjects (10% and 25%, respectively), despite the resulting mean fold change being >3. When comparing serum to SM/SLS (Figure 5B), among the metabolites exhibiting a mean fold change higher than 3 or lower than 0.5, Taurine (mean fold change >3) exhibited a reversed concentration ratio in 35% of the subjects. In both comparisons, the concentration ratios of all other metabolites remained consistent (> or <1) for all subjects. 

Metabolites with a mean fold change between 0.5 and 3 are listed in Table 1. The closer a mean fold change is to 1, the more likely its concentration ratios will vary around 1. For example, in the comparison of serum vs. SM/SLS, the mean fold change in Creatine was 1.5 (Table 1), but its concentration in serum was higher than in SM/SLS only in 12 out of 20 subjects. 

With a mean fold change <0.5 threshold, Acetate, Glutamate, and Creatine were significantly more concentrated in PS than in serum, while only Acetate was significantly more concentrated in SM/SLS than in serum (Figure 5A,B). 

Finally, the data indicate that the metabolites shared by the two major salivary gland samples were more concentrated in PS than SM/SLS (Appendix A).

#### 2.3.2. Metabolites Common to Serum and WS

Despite the complex origin of the WS, metabolite levels were compared using the same approach as for PS and SM/SLS. Statistical analysis showed that serum and WS concentrations differed significantly except for Choline and Tyrosine.

Based on Pearson analysis, 3-Hydroxybutyrate, Creatinine, Glutamate, and Lysine concentrations correlated with the corresponding serum concentrations with a coefficient >0.5 (Figure 6 and Table 1). 

Spearman correlation analysis confirmed the strong correlation of 3-Hydroxybutyrate and Creatinine found in Pearson analysis and assigned a moderate correlation (│r │≈ 0.5) to Lactate and Lysine.

Most of the mean fold changes (74%) fell between 1 and 20, with a few close to 1 (Figure 6 and Table 1). Among those metabolites with a mean fold change >3 or <0.5 (Figure 6), only Glycine, Proline, Pyroglutamate, Taurine, and Formate showed a few cases (10–30%) of a reversed concentration ratio.

Acetate and Glutamate were present in WS at a higher concentration than in serum for all study subjects. This is consistent with the result obtained when comparing serum and PS and partially in line with the comparison between serum and SM/SLS (Figure 5), where the amount of Glutamate was higher in SM/SLS than in serum in only 13 out of 20 subjects (0.9 mean fold change, Table 1). 

## 3. Discussion

Recent metabolomics studies have been performed to compare the diagnostic performance of serum and saliva or investigate their relationships in various pathologies [15,25,26,27,28]. Establishing concentrations and conditions for every analytical feature in healthy and pathological states is crucial for utilizing saliva in laboratory diagnostics. To our knowledge, this is the first metabolomics study to compare serum and salivary metabolite profiles obtained under physiological conditions.

We have already devised a standardized procedure for saliva collection, optimized the preparation of samples for ^1^H-NMR-based metabolomics, and generated the PS, SM/SLS, and WS metabolite profiles of young and healthy subjects [22,23]. Integrating these data with the corresponding serum metabolite profile allowed for new comparisons and more reliable clinical assessments. 

The mean metabolite concentrations obtained from the sera of the cohort of young and healthy subjects of this study were coherent with the data reported in The Human Metabolome Database (normal conditions, both sexes and age >18, https://hmdb.ca, accessed on 30 September 2023) [29], except for Glycerol and Urea. However, our saliva collection protocol requires a fasting period starting from midnight, which likely induces lipolysis. This process could cause an elevation in our serum Glycerol concentrations. As for the discrepancy in Urea content, we ascribe this fact to our experimental data acquisition conditions. The ^1^H-NMR Urea signal may have been affected by the exchange with water, leading to an underestimation of Urea serum concentration. 

Based on the total signal area distribution of the NMR spectra (*n* = 20), serum samples revealed a more compact distribution when compared with WS and PS samples [23], suggesting a minor individual variability in serum metabolite content compared with the salivary samples. 

In agreement with Tzimas and Pappa [30], we verified that human saliva and serum metabolomes are comparable in chemical composition but present significant differences in the concentrations of the common metabolites. In fact, in the pairwise comparisons of serum with PS, SM/SLS, or WS, the concentration distributions significantly differed for most metabolite features, with a few exceptions. 

By applying PLS-DA to the datasets, we found that VIP scores for component 1 effectively distinguished serum from saliva samples, while the three saliva types remained grouped, indicating quite comparable metabolomes in the salivary samples. Glucose and Lactate, whose concentrations were markedly higher in serum (Appendix A), and Acetate, which is predominantly derived from oral microbiota, are the principal compounds responsible for the observed separation.

The metabolite concentrations measured in serum and saliva samples revealed a low incidence of correlations in agreement with a previous study [26]. In serum vs. PS, 2-Hydroxyisovalerate, 2-Hydroxybutyrate, and 3-Hydroxybutyrate showed the highest correlation coefficients, and their presence in PS appeared to be supported by a favorable concentration gradient from serum to PS (Figure 5A, Appendix A). 2-Hydroxyisovalerate is derived from ketogenesis and branched-chain amino acid metabolism. Mammalian tissues (principally hepatic) that catabolize threonine (a ketogenic and glucogenic amino acid) produce 2-Hydroxybutyrate. 3-Hydroxybutyrate blood levels reflect fatty acid β-oxidation and ketogenic amino acid catabolism. The observed moderate to strong correlations relating to amino acid metabolism may arise from a transfer of 2-Hydroxyisovalerate, 2-Hydroxybutyrate, and 3-Hydroxybutyrate from serum to saliva. 

When comparing the serum and SM/SLS concentration datasets, Spearman’s correlation analysis assigned a moderately positive correlation (r ≈ 0.5) to 3-Hydroxybutyrate, in line with the serum vs. PS correlation analysis results. As this metabolite is more concentrated in serum than saliva, the parotid and submandibular/sublingual glands likely use the same molecular mechanism to transfer it from blood to saliva. Based on Pearson and Spearman analyses, 3-Hydroxybutyrate, Glutamate, Creatinine, and Lysine showed a positive correlation when comparing serum vs. WS. As for 3-Hydroxybutyrate, our data agree with the study of Miyazaki et al. [31]. We found a comparable correlation between serum and PS, suggesting that the parotid gland may be an entry point for that metabolite. Its increased concentration in the WS of patients with liver cirrhosis evidences its potential as a biomarker associated with the upregulation of the catabolic pathways of fatty acids/ketogenic amino acids [31]. However, we cannot exclude the contribution of oral microbiota in WS. In line with our findings, Jasim et al. found a correlation between Glutamate concentrations in stimulated WS and blood [32]. However, in that study, the concentration of Glutamate in WS was lower than that in serum, in contrast to our findings. Though we cannot exclude a contribution of oral microbiota metabolism, this apparent discrepancy may be due to the fact that glutamate salivary concentration declines with age [33], which may explain the higher salivary amounts measured in our young participants. The correlation between serum and WS Creatinine concentrations agrees with the literature [34], supporting the use of salivary Creatinine concentration as a non-invasive diagnostic tool for chronic kidney disease [35,36]. The literature does not support the correlation between serum and WS Lysine (essential amino acid) levels. 

Overall, for most of the metabolites shared by serum and saliva, we observed a prevalence of serum concentrations over the salivary ones (mean fold change >1, Figure 5 and Figure 6) without any significant correlation. In these instances, it might be inappropriate to attribute the levels of PS and SM/SLS metabolites exclusively to a transport mechanism from serum to saliva since the gland environment may have a role in producing and/or consuming these metabolites. 

The highest mean fold change values were obtained with serum vs. SM/SLS, indicating that saliva secreted from those glands is much more diluted than serum; the only exceptions were the Citrate, Creatine, Lactate, and Glucose mean fold changes obtained with serum vs. WS (Appendix A). When comparing the mean fold changes in serum vs. PS and serum vs. WS, the highest mean fold changes derived from the serum vs. WS, except for 2-Hydroxybutyrate, Glycerol, Glycine, Lysine, Phenylalanine, Proline, Pyruvate, and Tyrosine (Appendix A). The mean fold changes in the 12 amino acids in Appendix A indicate that the saliva secreted by SM/SLS has a lower aminoacidic content than PS and WS, and that PS has a lower Glycine, Lysine, Phenylalanine, Proline, and Tyrosine content than WS. These results suggest a potential use of amino acids for synthesizing secretory proteins in the parotid and submandibular/sublingual glands, whereas the proteolytic activity of the oral microbiota has a major influence on the amino acid content of WS. Nonetheless, it is plausible that differences in saliva flow and/or collection rate between the salivary types may have affected their metabolite content. Therefore, the prevalence of PS metabolite concentrations over SM/SLS (Appendix A) could be attributed to the lower salivary collection rate measured for PS (0.11 ± 0.08 vs. 0.20 ± 0.09 mL/min). We expect that the longer the saliva spends in the secretory acini of the gland, the more concentrated it may become. We calculated the fractional abundance of all amino acids in serum and saliva samples (Appendix A). Due to the different roles and origins of amino acids in blood and saliva, ratios of the same magnitude were observed only in limited cases (Histidine, Phenylalanine, and Leucine). Serum is remarkably rich in the gluconeogenic precursors Glutamine and Alanine, while Alanine, Glycine, Glutamine, and Glutamate are the most abundant amino acids in PS and SM/SLS. In WS, instead, the most abundant amino acids are Glycine, Glutamate, Lysine, and Proline. Notably, whilst amino acids undergo a complex homeostatic regulation resulting in stable plasma levels, salivary amino acid content is influenced by the proteolytic activity of oral microflora, which degrades salivary proteins and produces organic acids from amino acid fermentation. From this perspective, the high level of WS Proline can be explained by the degradation of salivary proline-rich proteins.

Notably, the mean concentration of Glucose in PS is significantly higher than in the other salivary samples (Appendix A), supporting the idea that parotid acini are the main route of entry for this metabolite [27]. However, the reduced collection rate of PS samples compared with SM/SLS and WS could have facilitated the accumulation of Glucose in PS. The same interpretation applies to PS Lactate, although this metabolite is also a product of the oral microbiota metabolism. We speculate that the Glucose concentration in WS is lower than that in PS due to its utilization as a microflora substrate [22]. 

However, our saliva-collecting protocol does not entirely exclude a certain degree of WS cross-contamination. Acetate, Glutamate, and Formate showed a mean fold change <1 in all saliva sample types (Figure 5 and Figure 6, Table 1 and Appendix A). The abundance of Acetate and Formate in WS is mainly due to the metabolism of oral microflora, and diffusion from WS could explain the presence of these metabolites in PS and SM/SLS. Notably, the mean concentrations of Acetate and Formate in saliva are associated with high standard deviation values (Appendix A), reflecting the variability in the composition of saliva microbiota. Glutamate is an important component of saliva because it contributes to taste [33], and it is the most abundant excitatory neurotransmitter in the central nervous system of vertebrates related to pain; its salivary level increases in patients with chronic migraine, validating its use as a clinical biomarker [37]. 

Finally, our evaluation revealed that some metabolites are unique to each sample type (Figure 3). We hypothesize that they represent the following: (a) in serum, metabolites characterized by a poor permeation of the hemato-salivary barrier; (b) in WS, intermediates of oral microbiota metabolism, shared by a cohort of subjects without any sign or symptom of oral/periodontal disease; and (c) in PS and/or SM/SLS, a group of endogenous metabolites produced by acinar and/or ductal cells of the major salivary glands. Further investigations in a larger population are necessary to confirm these hypotheses.

## 4. Materials and Methods

According to the ethical principles of the Declaration of Helsinki, written informed consent was obtained from all the study participants. 

As described in reference [22], the cohort of participants consisted of twenty healthy volunteers aged between 20 and 25 years (Appendix A). Subjects were eligible for saliva collection after completing an oral clinical examination and a sialometry test to exclude dental/periodontal disease or hyposalivation. Health status was checked with a medical history interview. The metabolite profiling of WS, PS, and SM/SLS samples was the topic of a previous publication wherein the study population, the saliva collection procedure, the saliva sample preparation, and the ^1^H-NMR spectra collection and analysis are described [22]. Briefly, saliva and blood collection took place from 8 to 10 a.m. The participants were instructed to refrain from eating, smoking, and performing intense physical activity for at least 12 h before sampling. They were also asked to refrain from oral hygiene practices for 45 min before saliva collection. Immediately before saliva collection, patients rinsed their mouths with water for 60 s. PS, SM/SLS, and WS were collected separately under unstimulated conditions. To collect PS and SM/SLS, a sterile sponge was positioned over the duct outlets of the glands to absorb the secreted saliva. The sponge was squeezed with a syringe to collect saliva in a vial at regular intervals. WS was collected using the passive drooling method. The serum metabolite profile derived from the subjects and its comparison with saliva profiles from the same subjects is the topic of this manuscript. 

### 4.1. Blood Collection, Serum Sample Preparation, and Processing

After saliva collection and under fasting conditions, 9 mL of cubital venous blood was collected from all participants. Blood samples were kept at 37 °C for 30 min in tubes without anticoagulant and then centrifuged at 4000× *g* for 10 min at 25 °C to separate serum. The procedure resulted in 20 serum samples ranging in size from 2.0 to 3.8 mL. Serum samples were stored at −80 °C until metabolomic analysis.

For NMR sample preparation, sera were thawed at room temperature and, since the presence of plasma proteins may interfere with metabolite quantification via NMR [38], they were protein-depleted via ultrafiltration using centrifugal filters (3000 MWCO, Amicon Ultra-4 Centrifugal filters, Merck Millipore, Darmstadt, Germany) at 4000× *g* for 120 min at 10 °C. Each serum ultrafiltrate was used to prepare a 600 µL sample, including the addition of 10 µL of 1 M phosphate buffer (pH 7.4) and 15 µL of 1% 3-trimethylsilyl propanoic acid (TSP) in 2.5% D_2_O (as a quantitative internal standard).

One-dimensional ^1^H-NMR spectra of serum samples were acquired at 25 °C with a JEOL 600 MHz ECZ600R spectrometer (JEOL Inc., Tokyo, Japan) as described in [39]. The spectra were processed and analyzed with the Chenomx NMR suite 9.0 software (Chenomx Inc., Edmonton, AB, Canada), zero-filling to 256 K points and using a line broadening of 0.5 Hz.

### 4.2. Metabolomics Data Analysis

Multivariate statistical analysis was conducted on target metabolites using Metaboanalyst 5.0 (https://www.metaboanalyst.ca, accessed on 30 September 2023) [40]. Partial least-squares–discriminant analysis (PLS-DA) was used to compare the original metabolite composition of all sample matrices. The PLS-DA results were visualized as 2D score plots and VIP scores [41]. VIP scores ≥2 were considered relevant for the PLS-DA analysis.

### 4.3. Statistical Analysis

Data were analyzed using Origin 2019 software. Descriptive statistics for each sample type and variable are presented as mean  ±  SD or median and interquartile range. The mean fold change was obtained for each metabolite by averaging the ratios calculated by dividing the serum concentration by the PS, SM/SLS, or WS concentration of all study subjects (*n* = 20).

Differences between concentration distributions of two sample types were evaluated using the Mann–Whitney test, a non-parametric statistical test used to compare two samples or groups. 

The Pearson and Spearman correlation tests were performed on the observed concentrations of shared metabolites to explore the relationships between serum and saliva composition. The correlation coefficient (r) ranges from +1 (perfect positive correlation) to –1 (perfect negative correlation). For absolute values of r, the correlation is considered strong if 0.6 < r < 0.8, moderate if 0.4 < r < 0.6, and weak if 0.2 < r < 0.4. 

For all analyses, the significance level was set at *p* < 0.05.

Metabolites with concentrations below the quantification limit [23] in ≥20% of participants were excluded from all analyses.

## 5. Conclusions

Serum and salivary metabolite profiles derived from samples collected concurrently from healthy subjects under physiological conditions were compared to assess the relationships between their compositions. Many researchers have focused exclusively on WS, especially when searching for biomarkers, because it can be collected using non-invasive procedures and is rich in many analytes of interest for diagnosis or screening. Instead, we intended to study the three types of saliva separately to provide a comprehensive analysis of their metabolomes in relation to serum composition under physiological conditions. Our results highlight that none of the saliva types are a mere ultrafiltrate of blood, as saliva composition is influenced by several factors, such as intrinsic glandular metabolism and the enzymatic activities of the oral microbiota. The proximity of the capillary bed and acinar/ductal structures suggests that metabolites may diffuse or be transported from serum to saliva because of a favorable concentration gradient and/or via specific transport systems. Based on our analyses, however, an intense oral metabolism is likely to alter the concentrations of most excreted metabolites, both at the gland level and in other oral niches, thus preventing very strong correlations between serum and salivary concentrations. In some cases, the salivary metabolite was not detected in the serum, or its concentration exceeded the serum level, indicating intense endogenous production. 

We did find some moderate to strong correlations between the serum and saliva concentrations of specific metabolites, supporting the idea that these metabolites may have been transferred from serum to saliva to an extent dependent on their concentrations. Our results align with the literature focused on those metabolites. 

In terms of future applications, the metabolomic profiles of serum, WS, PS, and SM/SLS generated using standard operating procedures and the resulting assessments may contribute to a valuable knowledge base for salivary metabolomics aimed at identifying biological markers of oral and systemic health disorders [7,30].

## Figures and Tables

**Figure 1 ijms-24-16603-f001:**
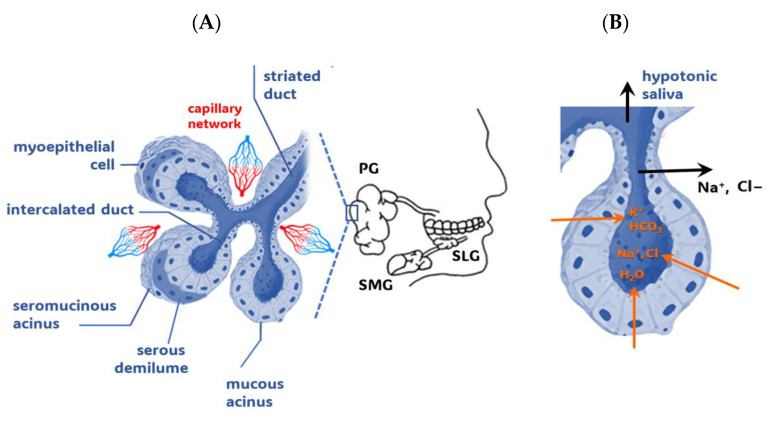
Salivary gland components. (**A**) Detail of a mixed salivary gland showing one mucous acinus and two seromucinous acini, the ductal system, and the capillary network. Contractile myoepithelial cells wrap around the acinar cells and their ducts to promote gland secretion. In mixed seromucinous acini, the serous cells form a demilune around the mucous acinus. Each acinar portion merges into an intercalated duct, which continues as a striated duct. (**B**) An acinus with its intercalated duct. According to the current model of saliva secretion, in the first step, the acinar cells secrete a relevant number of Na^+^ and Cl^−^ ions into the acinar lumen. This ionic flux drives water efflux transcellularly (through aquaporin channels) and paracellularly, producing an isotonic fluid that enters the ductal lumen. In the second step, the ductal cells reabsorb most of the Na^+^ and Cl^−^ and secrete K^+^ and HCO_3_^−^ ions, resulting in hypotonic saliva reaching the final secretory duct [4,9,10]. PG, parotid gland; SMG, submandibular gland; SLG, sublingual gland.

**Figure 2 ijms-24-16603-f002:**
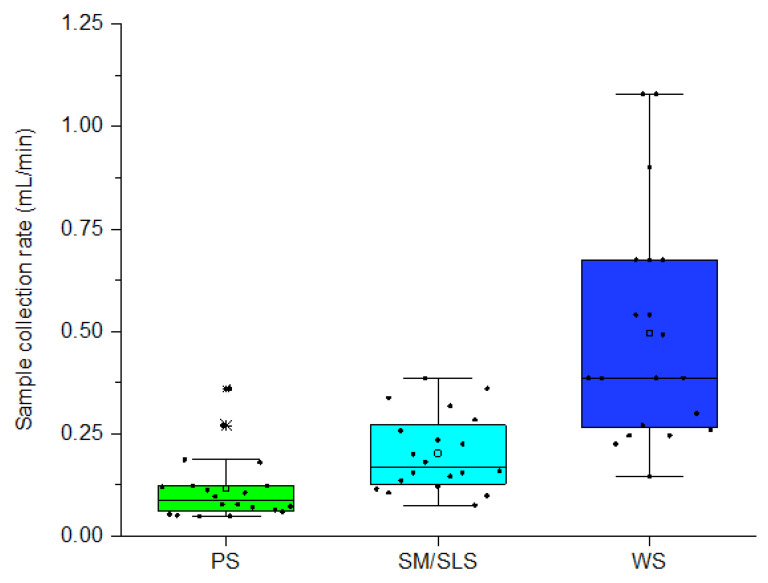
Collection rates of the salivary samples. The boxes are determined by the 25th and 75th percentiles; the horizontal line within each box is the median of the group, while the empty square inside the box corresponds to the mean value of the group; the asterisks identify two outliers. Black dots represent individual sample collection rates.

**Figure 3 ijms-24-16603-f003:**
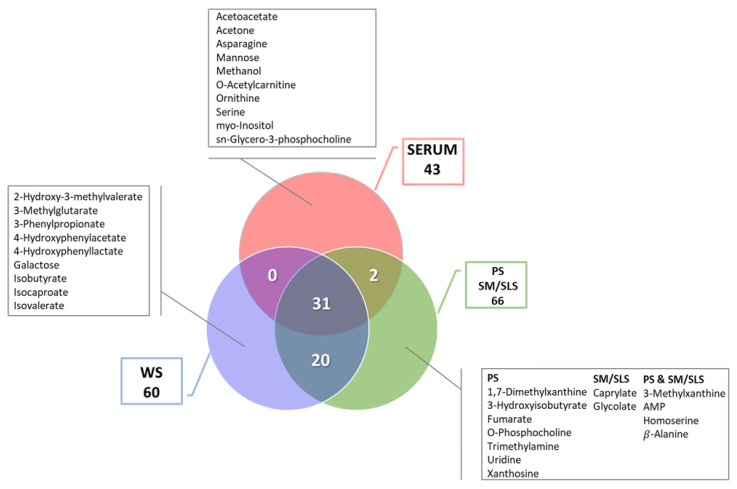
Venn diagram of the number of metabolites detected in each sample type. The red, blue, and green circles refer to the metabolite profile of serum, whole saliva (WS), and saliva produced by parotid and submandibular/sublingual glands (PS and SM/SLS), respectively. Overlapping regions reveal the number of shared metabolites. Unique metabolites of each sample are framed in gray squares. For PS and SM/SLS samples, exclusive metabolites are reported as present only in PS, only in SM/SLS, and both in PS and SM/SLS. For each metabolite profile, the total number of metabolites can be calculated by adding the number of shared and unique metabolites.

**Figure 4 ijms-24-16603-f004:**
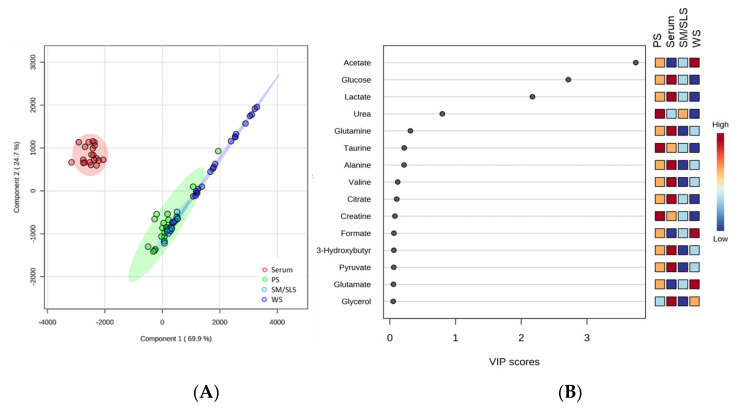
Partial least-squares–discriminant analysis (PLS-DA) based on the metabolites common to serum, WS, and PS plus SM/SLS profiles. (**A**) PLS-DA 2D scores plot. (**B**) Metabolite ranking (top 15 metabolites) according to the Variable Importance in Projection (VIP) scores, resulting from the separation by component 1 in the PLS-DA score plot. A higher metabolite VIP score denotes a more significant contribution to sample separation. The colored boxes on the right indicate the relative metabolite abundance in each sample type.

**Figure 5 ijms-24-16603-f005:**
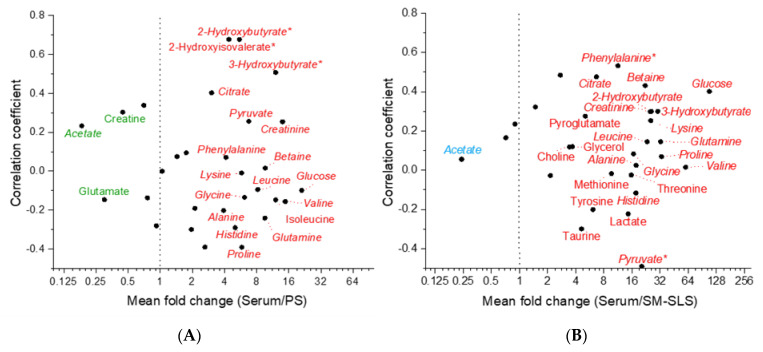
Mean fold change in the metabolites shared by serum and parotid saliva (**A**) or submandibular/sublingual saliva (**B**) expressed as a function of their Pearson’s correlation coefficients. For each metabolite, the mean fold change was obtained by averaging the ratios obtained by dividing the concentration in serum by the concentration in the PS or SM/SLS of the 20 study subjects. Line x = 1 marks the boundary between two graph subsections: on the left, the salivary concentrations are expected to be higher than the serum concentration; on the right, the salivary concentrations are expected to be lower than the serum. Metabolites with a mean fold change lower than 0.5 and higher than 3 are labeled. They appear in green or cyan when all their original metabolite concentrations are higher in PS or SM/SLS than in serum, respectively, and in red when their original metabolite concentrations are higher in serum than in PS or SM/SLS. Metabolites in *italics* appear in the same subsection of the two graphs. The asterisks mark the metabolites with a correlation coefficient ≥±0.5. X axis is in log_2_ scale.

**Figure 6 ijms-24-16603-f006:**
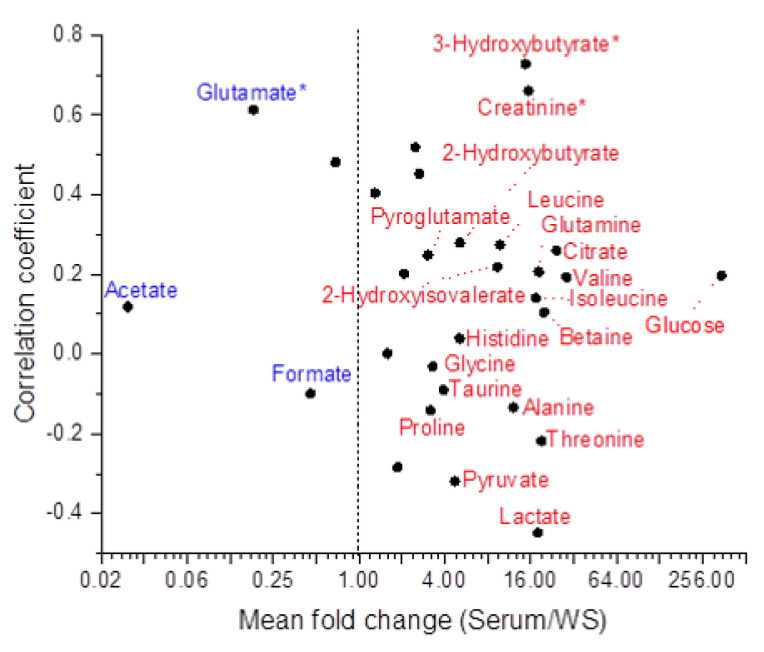
Mean fold changes in the metabolites shared by serum and WS are expressed as a function of their correlation coefficients. The line x = 1 marks the boundary between two graph subsections: on the left, the salivary concentrations are expected to be higher than the serum concentrations; on the right, the serum concentrations are expected to be higher than the salivary concentrations. Metabolites with mean fold changes <0.5 and >3 are labeled. They appear in blue when all their original metabolite concentrations are higher in WS than in serum and red when they are higher in serum than in WS. Urea was omitted as it was not found in several WS samples. The asterisks mark the metabolites with a correlation coefficient >0.5. X axis is in log_2_ scale.

**Table 1 ijms-24-16603-t001:** Metabolites with a mean fold change between 0.5 and 3.

Metabolite	SERUM vs. PS Mean Fold Change	Metabolite	SERUM vs. SM/SLS Mean Fold Change	Metabolite	SERUM vs. WS Mean Fold Change
Aspartate	0.7	Formate	0.7	Aspartate	0.7
Formate	0.8	Glutamate *	0.9	Choline *	1.3
Taurine	0.9	Creatine *	1.5	Glycerol	1.6
Choline *	1.0	Urea *	2.1	Tyrosine *	1.9
Urea	1.4	Aspartate	2.7	Creatine	2.1
Glycerol	1.7			Lysine ^a^	2.5
Tyrosine	1.9			Phenylalanine	2.6
Arginine	2.1				
Lactate	2.6				

* Serum and salivary distributions of the observed concentrations did not significantly differ at *p* = 0.05. ^a^ Metabolite with a correlation coefficient of 0.52.

## Data Availability

Data is contained within the article or Appendix A.

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
