# Peer review of "Human Serum and Salivary Metabolomes: Diversity and Closeness"

_ijms, 2023, doi:10.3390/ijms242316603_

Round 1

Reviewer 1 Report

Comments and Suggestions for Authors

A review of the manuscript titled „The human serum and salivary metabolomes: diversity and closeness” by Elena Ferrari, Mariana Gallo, Alberto Spisni, Rita Antonelli, Marco Meleti, Thelma A. Pertinhez to Int. J. Mol. Sci.

The manuscript includes a comparison of selected metabolites determined in serum and parotid saliva (PS), submandibular/sublingual saliva (SM/SLS), and WS samples. The test is interesting and gives an insight into how saliva can be collected non-invasively, easily, and quickly. The authors were fully critical of the obtained results, stating that despite many advantages, saliva will not replace serum in metabolomics analysis. Please reconsider the following:

1.      How were parotid saliva (PS), submandibular/sublingual saliva (SM/SLS), and WS samples obtained? Such information should be included in the Methods section.

2.      In Figure 2, the sentence "the line and the square inside the box are the median and the mean, respectively" is not entirely understandable. The results are presented as median or mean? This should be determined by the result of the distribution.

3.      If some of the results had a non-parametric distribution (presented as median), why was Pearson's correlation used?

4.      I suggest replacing the "i, ii, iii" scores with "a, b, c".

Reviewer 2 Report

Comments and Suggestions for Authors

1. Nothing is said about the study group of volunteers. Gender, age are not indicated, there is no information about smoking status, oral health, presence/absence of dental diseases, etc. All these factors are significant in saliva analysis and may cause significant differences from the blood plasma metabolome.

2. The separation of groups by the method of principal components is not a new discovery, but it is natural that in the presence of indicators whose concentrations in plasma and saliva differ by orders of magnitude (for example, glucose, lactate), the plasma group will be separated from the rest. It would be interesting to see a comparison of groups using this method if we exclude from the analysis components with VIP scores > 2. What parameters then will primarily determine the division? Similarly, due to the higher content of total protein (70 and 1 g/l on average for plasma and saliva), the content of amino acids in plasma will be higher. How accurate is this comparison? It might be more logical to consider how the relative content of these components changes (for example, normalized to the content of the same protein)?

3. The authors use several saliva exudates: parotid saliva (PS), submandibular/sublingual saliva (SM/SLS), and WS. Which of these types is more similar in composition to blood plasma? At what stage do the differences between blood and saliva increase? Why were separate PS and SM/SLS needed if in practice all researchers work with whole saliva? In conclusion, I didn’t see these moments.

Round 2

Reviewer 2 Report

Comments and Suggestions for Authors

The authors gave comprehensive answers to the reviewer’s remarks and comments and made appropriate changes to the text of the article. I believe that in its present form the manuscript can be recommended for publication.